# Hair Follicle Targeting and Dermal Drug Delivery with Curcumin Drug Nanocrystals—Essential Influence of Excipients

**DOI:** 10.3390/nano10112323

**Published:** 2020-11-23

**Authors:** Olga Pelikh, Cornelia M. Keck

**Affiliations:** Department of Pharmaceutics and Biopharmaceutics, Philipps-Universität Marburg, Robert-Koch-Str. 4, 35037 Marburg, Germany; olga.pelikh@pharmazie.uni-marburg.de

**Keywords:** curcumin, poor solubility, dermal drug delivery, image analysis, skin, hair follicle targeting, nanocrystals, nanosuspension

## Abstract

Many active pharmaceutical ingredients (API) possess poor aqueous solubility and thus lead to poor bioavailability upon oral administration and topical application. Nanocrystals have a well-established, universal formulation approach to overcome poor solubility. Various nanocrystal-based products have entered the market for oral application. However, their use in dermal formulations is relatively novel. Previous studies confirmed that nanocrystals are a superior formulation principle to improve the dermal penetration of poorly soluble API. Other studies showed that nanocrystals can also be used to target the hair follicles where they create a drug depot, enabling long acting drug therapy with only one application. Very recent studies show that also the vehicle in which the nanocrystals are incorporated can have a tremendous influence on the pathway of the API and the nanocrystals. In order to elucidate the influence of the excipient in more detail, a systematic study was conducted to investigate the influence of excipients on the penetration efficacy of the formulated API and the pathway of nanocrystals upon dermal application. Results showed that already small quantities of excipients can strongly affect the passive dermal penetration of curcumin and the hair follicle targeting of curcumin nanocrystals. The addition of 2% ethanol promoted hair follicle targeting of nanocrystals and hampered passive diffusion into the stratum corneum of the API, whereas the addition of glycerol hampered hair follicle targeting and promoted passive diffusion. Propylene glycol was found to promote both pathways. In fact, the study proved that formulating nanocrystals to improve the bioefficacy of poorly soluble API upon dermal application is highly effective. However, this is only true, if the correct excipient is selected for the formulation of the vehicle. The study also showed that excipients can be used to allow for a targeted dermal drug delivery, which enables to control if API should be delivered via passive diffusion and/or as drug reservoir by depositing API in the hair follicles.

## 1. Introduction

Besides oral drug delivery, topical drug delivery is a major and constantly growing market in the pharmaceutical field. The compound annual growth rate (CAGR) is >5% and the global market is projected to reach 123.2 billion U$ by 2024 [1]. The high prevalence of skin diseases or injuries, as well as the increasing demand of topically applied drugs for transdermal drug delivery, require the development of innovative and effective formulations that can overcome current challenges in this field. Challenges that need to be addressed are for example a sufficient penetration of the active compound into or through the skin without skin irritation, e.g., due to the use of penetration enhancers. Ideally, the formulation should not only deliver the active compound but should also help to maintain or restore the skin barrier function. The therapy concept that combines efficient drug delivery and skin protecting or restoring properties at the same time, is known as advanced corneotherapy and can help to improve the treatment efficacy and compliance of the patients [2]. In addition, there is an increasing demand to formulate topical drug products as a depot, i.e., formulations that are applied once and then release the active compound over an extended period of time. Most prominent examples for extended dermal drug delivery are transdermal patches that can be worn for several days or microneedle patches where biodegradable drug-loaded microneedles are deposited into the skin where they can act as a long-term reservoir for drugs [3,4,5].

Another method to form drug depots for long-term drug release is the delivery of active compounds via the hair follicles. Hair follicle targeting gained much attention in the last years and is now known to be most efficient if the drug is encapsulated or linked to a nanocarrier that preferentially possesses a size of about 400–700 nm [6,7,8,9,10]. The improved penetration of active compounds in particulate form is explained by the ratchet effect. This means that the cuticular hair structure acts as a ratchet and transports the particles deeper into the hair follicle [11]. Once the drug-loaded carrier has reached the hair follicle, it will remain for up to several days, forming a long-lasting drug reservoir from which drug is constantly released. Due to the special anatomy of the hair follicle stratum corneum, which is less developed and thus more permeable for chemical compounds compared to the stratum corneum of the skin, the drugs can diffuse transfollicularly and directly into the viable layers of the skin. The possibility to combine long-lasting drug depots and efficient drug delivery into the viable dermis at the same time and without the need of barrier disruption or the use of skin irritant excipients, makes drug delivery via the follicular route highly attractive.

However, even though the superiority of nanocarriers for hair follicle targeting is now well-known, an efficient application of nanocarriers as drug product will always require the formulation of such carriers in appropriate vehicles. Vehicles can be hydrophilic, lipophilic and can vary in viscosity, polarity and many other parameters. When looking at the passive, dermal penetration of chemical compounds, it is well-known that different vehicles with different properties can strongly affect the penetration properties of active compounds into the skin. Consequently, it can be assumed that vehicles with different properties might also affect the penetration of drug nanocarriers into hair follicles. A recent study already addressed this issue and investigated the influence of the type of vehicle on the penetration efficacy of curcumin nanocrystals into the hair follicles. In this study nanocrystals were incorporated into different types of gels that varied in polarity, lipophilicity and viscosity [12]. Results demonstrated that the penetration efficacy of the curcumin nanocrystals into the hair follicles was not affected by variations in viscosity, polarity or lipophilicity of the vehicle. In contrast, it was found that the variation in vehicle properties led to tremendous differences in the passive, dermal penetration of curcumin from the drug nanocrystals. The most influencing parameter was the ability of the vehicle to hydrate the stratum corneum. Hence, vehicles with good skin-hydrating properties led to the best dermal penetration of the curcumin. To substantiate this finding, a humectant was added to a formulation with poor hydrating properties and to a formulation with good skin hydrating properties, respectively. Results could nicely demonstrate the increase in skin hydration upon the addition of the humectant to the formulation with former poor skin hydration. And, as expected, the increase in skin hydration resulted in a pronounced increase in passive, dermal diffusion of curcumin. In contrast, the addition of the humectant to the formulation that already showed good skin hydration could not further increase the skin hydration. Consequently, the passive dermal diffusion of curcumin through the skin was not much increased.

Besides, it was also observed, that the addition of the humectant reduced the follicular uptake of the curcumin nanocrystals by about 20%. Based on obtained results, it was hypothesized that a swelling of the hair shaft and/or changes in the structure of the hair cuticula might be responsible for this. The changes might cause a reduced ratchet effect, which could than result in a less efficient transport of the nanoparticles into the hair follicle. Therefore, the aim of this study was to investigate and to understand this observation in more detail. For this, different excipients with and without known skin-hydrating properties were selected. Subsequently, curcumin nanocrystals were produced after a previously established protocol [12,13], and the different excipients were added in different concentrations to the nanosuspension. The obtained nanocrystal formulations with and without excipients were first characterized regarding size and size distribution and were then investigated via the ex-vivo porcine ear model regarding passive, dermal diffusion of curcumin, their influence on the biophysical properties of the skin (i.e., skin hydration, barrier function and skin roughness) and their ability to transport curcumin nanocrystals into the hair follicles (i.e., hair follicle targeting).

The present study is aimed to identify excipients that are able to promote hair follicle targeting and/or passive diffusion of curcumin through the skin. Based on these findings it is then possible to optimally formulate drug nanocrystals for a targeted dermal drug delivery. Hence, with purposeful selection of excipients, one can choose if the active compound should be delivered via passive diffusion and/or as long-lasting drug reservoir by depositing drug nanocrystals within the hair follicles.

## 2. Materials and Methods

### 2.1. Materials

Curcumin was used as the model drug in this study. It is a poorly water-soluble active substance (BCS class IV) that can be detected in the skin and in the hair follicles due to its pronounced autofluorescence, which enables the visualization of penetrated active from the different formulations. It was obtained from Receptura Apotheke-Cornelius Apothekenbetriebs OHG (Frankfurt am Main, Germany). TPGS (d-α-tocopherol polyethylene glycol 1000 succinate) was used as stabilizer for the nanocrystals and was purchased from Gustav Parmentier GmbH (Frankfurt am Main, Germany). Purified water was obtained from a PURELAB Flex 2 (ELGA LabWater, High Wycombe, UK) and was used as a liquid phase for the nanocrystals. The excipients used were glycerol (extra pure, Acros Organics, Geel, Belgium), urea (Caesar & Loretz GmbH, Hilden, Germany), propylene glycol (Caesar & Loretz GmbH, Hilden, Germany), ethanol (HPLC grade, Fischer Scientific GmbH, Schwerte, Germany), and olive oil (Gustav Heess GmbH, Leonberg, Germany). All chemicals were used as received.

### 2.2. Methods

#### 2.2.1. Production and Characterization of Curcumin Nanocrystals with Different Excipients

Nanocrystals were produced by using the small-scale bead milling approach [14] and by using a previously established protocol [12,13]. The nano-milling was carried out in an 8 mL brown glass bottle filled with Yttrium stabilized zirconium oxide beads (0.3–0.4 mm, SiLibeads^®^, Sigmund Lindner GmbH, Warmensteinach, Germany) and equipped with a magnetic stirrer (Ø: 8 mm, 13 mm, Rotilabo^®^, Carl Roth GmbH + Co. KG, Karlsruhe, Germany). The bead/suspension-ratio was 60/40 (*v*/*v*) and the coarse bulk suspensions contained 5.0% (*w*/*w*) curcumin, 1.0% (*w*/*w*) TPGS and water to up to 100% (*w*/*w*). The closed bottle was fixed horizontally to a magnetic stirrer (MIXdrive 15, 2mag AG, Munich, Germany) with adhesive tapes and the milling process was performed at 1000 rpm for 24h. The so obtained nanocrystals were allocated and different excipients with different concentrations were added to these aliquots (Table 1). Subsequently, the formulations were characterized regarding their size and size distribution by using three independent techniques, i.e., dynamic light scattering (DLS, Zetasizer Nano ZS, Malvern Panalytical Ltd., Malvern, UK), laser diffraction (LD, Mastersizer 3000, Malvern Panalytical Ltd., Malvern, UK) and light microscopy (Olympus BX53 equipped with an Olympus SC50 CMOS color camera, Olympus soft imaging solutions GmbH, Warmensteinach, Germany). DLS data were analyzed with the general purpose mode and LD data analysis was done with Mie-theory with optical parameters set to 1.87 for the real refractive index and 0.1/0.01 for the blue light (470 nm) and the red light (633 nm) imaginary refractive indices, respectively.

#### 2.2.2. Ex-Vivo Model for Passive Dermal and Follicular Penetration

The influence of excipients on the passive, dermal penetration of curcumin and the influence on the follicular penetration of the curcumin nanocrystals was tested by utilizing the ex-vivo porcine ear skin model [15]. For this, fresh porcine ears were obtained from a local slaughterhouse and used within a few hours after butchering. Prior to the application of the nanosuspensions on the ventral side of the ear, ears were washed with lukewarm water (approx. 23–25 °C), dried with paper towels and fixed to flat polystyrene plates that were covered with aluminium foil. Intact skin areas of 2 × 2 cm^2^ without wounds and scratches were selected and marked. The hair within these areas was cut to a length of about 1–3 mm. On each test area, 50 µL formulation were applied and massaged in for 3 min with a saturated, gloved finger [12]. Formulations were allowed to penetrate into the skin and the hair follicles for 6 h in an oven with a temperature set to 32 °C. After incubation, the formulations were carefully washed off and punch biopsies (15 mm drive punch) were taken from each test area. The skin biopsies were immediately embedded (Tissue-Tek^®^ O.C.T.™, Sakura Finetek Europe B.V., Alphen aan den Rijn, The Netherlands), frozen and stored at −20 °C until further use. Experiments were performed in triplicate and each formulation was tested on three different ears.

##### Determination of Follicular and Dermal Penetration via Epifluorescence Microscopy

Curcumin possesses a strong autofluorescence. Therefore, epifluorescence microscopy can be used to detect curcumin and curcumin nanocrystals that penetrated either into the skin and/or into the hair follicles [12]. Prior to microscopic analysis, 40 μm thick skin sections were prepared with a cryomicrotome (Frigocut 2700, Reichert-Junk, Nußloch, Germany). The cutting was carried out from right to left in order to avoid contamination with the curcumin from the skin surface. The skin cuts were then subjected to inverted epifluorescence microscopy (Olympus CKX53 equipped with an Olympus DP22 color camera, Olympus Deutschland GmbH, Hamburg, Germany). The intensity of the fluorescence light source (130 W U-HGLGPS illumination system, Olympus Deutschland GmbH, Hamburg, Germany) was kept constant and was set to 50% and 25% for the skin and the hair follicle sections, respectively. Also, the exposure time was kept constant and was set to 50 ms for all images taken. The fluorescence filter used was the DAPI HC filter block system (excitation filter: 340–390 nm, dichroic mirror: 410 nm, emission filter: 420 nm (LP)).

##### Digital Image Analysis

Digital image analysis was used to measure the mean penetration depth (MPD) of curcumin into and through the skin as well as the penetration depth of the curcumin nanocrystals into the hair follicles. In addition, it was used to quantify the mean grey value (MGV) of each image as a surrogate for the total amount of penetrated curcumin into the skin [16]. In addition, changes in stratum corneum thickness caused due to the treatment of the skin with the different formulations were also analyzed.

The passive, dermal penetration efficacy of curcumin from the different formulations, i.e., MPD and MGV, were determined according to a previously established protocol [16] with ImageJ software [17,18,19]. For this, all images were subjected to an automated threshold protocol in order to eliminate the autofluorescence of the skin. Consequently, the remaining fluorescence intensity (MGV) within the images corresponded to the penetrated amount of curcumin and was used as semi-quantitative surrogate for the total amount of penetrated curcumin. The MPD corresponds to the mean penetration depth that was determined by measuring the maximal penetration depth of curcumin from each image. The mean thickness of the stratum corneum was assessed similarly. For the measurements, 36 images (12 images from each ear, 3 ears in total) were analyzed for each formulation. The penetration depth of the nanocrystals into the hair follicles was determined on 5–10 follicles per porcine ear and on 3 different ears for each formulation. The measurements were performed with the cellSens Entry software package (Olympus Cooperation, Tokyo, Japan).

#### 2.2.3. Determination of Biophysical Skin Properties

The biophysical properties of the skin after the application of the different formulations were determined in order to investigate the effect of the different excipients on the barrier function, skin hydration and skin friction. The transepidermal water loss (TEWL) is a measure for the amount of water that passively evaporates through the skin. It is therefore frequently used to characterize the skin barrier function [20] and was measured with a Tewameter (Tewameter^®^ TM300, Courage & Khazaka electronic GmbH, Cologne, Germany). The skin hydration was assessed by measuring the capacitance of the skin surface (Corneometer^®^ CM825, Courage & Khazaka electronic GmbH, Cologne, Germany), where a high capacitance represents a high hydration level of the skin and vice versa. The skin friction is the force needed to rotate a small disc on the skin surface and was measured by using a frictiometer (Frictiometer^®^ FR700, Courage & Khazaka electronic GmbH, Cologne, Germany). The influence of the excipients on TEWL, skin hydration and skin friction was assessed on fresh pig ears that were conditioned and treated as described above. The biophysical skin properties were assessed on each test area prior to application of the formulations and after a penetration time of 6 h. Untreated skin areas that possessed TEWL values >13 indicated an impaired barrier function and thus were excluded from the study. Each formulation was tested in triplicate on three different ears.

#### 2.2.4. Statistical Analysis

Descriptive statistics and the comparisons of the mean values were analyzed with JASP software (version 0.13.1) [21] and Minitab 19 (Minitab Inc., State College, PA, USA), respectively. Parametric data were subjected to a one-way ANOVA, that was Welch-adopted in case of variance heterogeneity. Tukey post-hoc and Games–Howell post-hoc tests were performed to compare the mean values. For the nonparametric data, Kruskal–Wallis analyses of variance with Dunn‘s post-hoc tests were performed [22]. In case of multiple comparisons, the Bonferroni–Holm adjustment was applied to avoid alpha error accumulation. *p* -values < 0.05 were considered to be statistically significant.

## 3. Results and Discussion

### 3.1. Production and Characterization of Curcumin Nanocrystals with Different Excipients

The coarse bulk material contained curcumin crystals in the upper micrometer range (Table 2). The small-scale bead milled curcumin nanosuspension (NS d1) possessed a size of about 250 nm and a polydispersity index (PDI) of 0.25 (Table 2). Thus, possessing similar properties than the previously produced curcumin nanocrystals [12,13]. Light microscopy and laser diffraction revealed the presence of some larger curcumin microcrystals (approx. 4–10 µm), which were not destroyed during the milling process. The larger crystals caused some physical instability, because the broad size distribution of the nanosuspension initiated Ostwald ripening during storage for 14 days (NS d14) at room temperature. During this time, the z-average and the PDI almost doubled, leading to sizes of about 500 nm and to a broad size distribution (PDI > 0.4). Particle growth due to Ostwald ripening was also confirmed by laser diffraction. However, LD and light microscopy could also confirm that, despite a moderate increase in the mean size, no larger aggregates were formed during storage (Table 2, Figure 1).

The addition of excipients caused small changes in the size and size distribution (Table 2, Figure 1). Reasons for these changes are the increased solubility of curcumin in these excipients. The addition of excipients to the original nanosuspension leads to a partial dissolution of the nanocrystals, which can result in either larger or lower particle sizes (c.f. Table 2, LD data), when compared to the nanosuspension without excipients [23,24]. Additives that cause only a minor increase in solubility will decrease the particle size, because only small amounts of curcumin are dissolved from the surface of each particle. In contrast, if the excipients are good solubility enhancers, they cause a pronounced dissolution of curcumin. As small particles possess a higher kinetic solubility and increased dissolution velocity compared to larger sized crystals, the small crystals will dissolve first and will leave the larger particles behind. This will then lead to an all-over increased particle size of the whole formulation.

The solubility of curcumin is very good in ethanol and propylene glycol [25] and the largest particles were thus found for the formulations to which either ethanol or propylene glycol were added (DLS data, Table 2, Figure 1). The addition of a small quantity of olive oil resulted in the formation of some oil droplets within the aqueous phase of the nanosuspension and some of the curcumin nanocrystals even adhered to the surface of these droplets. Consequently, from all the formulations produced, the addition of olive oil resulted in the formulation with the largest measured particle size and the broadest size distribution, respectively (LD data, d(v) 0.9–0.99 values, Table 2, Figure 1).

In a previous study by Pelikh et al., the particle size of nanocrystals was shown to influence the passive diffusion of hesperetin from drug nanocrystals [26]. However, a tremendous increase in passive diffusion was only achieved with particle sizes <200 nm. Larger sizes did not show pronounced differences. In addition, it was found that the effect decreased with increasing penetration depth into the skin and was already cancelled out after 10 tape strips, which corresponds to less than half of the thickness of the stratum corneum. The present study aims at investigating not only the penetration of curcumin into the upper layers of the stratum corneum but also into the deeper layers of the skin. All nanocrystals used in this study possessed a size well above 200 nm and therefore the differences in size—caused by a partial dissolution of the nanocrystals upon the addition of the solvents—were considered to affect the passive dermal penetration only to a minor extend.

### 3.2. Determination of Passive Dermal Penetration

#### 3.2.1. Microscopic Imaging by Inverted Epifluorescence Microscopy

The passive diffusion of curcumin was found to be strongly influenced by the addition of the different excipients to the original curcumin nanosuspension (Figure 2). Especially the addition of 5% glycerol or 5% propylene glycol seemed to increase the penetration efficacy of curcumin. The addition of these excipients increased the amount of curcumin that penetrated into the stratum corneum and enabled a transdermal penetration of the curcumin into the viable dermis. A penetration enhancing effect, even though it was not as pronounced as for the above-mentioned excipients, was also detected for glycerol 2% and urea 5%. No differences compared to the nanosuspension without additives were found for the formulations that contained olive oil or 10% urea. For ethanol even a slightly decreased penetration was observed. In addition to affecting the penetration of curcumin, the additives were also found to affect the thickness of the stratum corneum (SC). When compared to untreated skin, the SC seemed to be thicker after the application of the nanosuspensions. The effect seemed most pronounced for the nanosuspensions that contained 5% glycerol or 5% propylene glycol. In contrast, the addition of ethanol seemed to prevent the swelling of the SC. The increase or decrease in SC thickness correlates with the amount of penetrated curcumin. Hence, data indicate, that excipients that increase the SC thickness can improve the penetration efficacy of the hydrophobic curcumin, whereas excipients that cannot increase the SC thickness hamper the penetration.

#### 3.2.2. Digital Image Analysis

Data obtained from the visual inspection of the images enabled a first impression of the influence of the different excipients on the dermal penetration efficacy of curcumin from nanocrystals. However, a clear picture can only be obtained by subjecting the images to a digital image analysis which enables the transfer of the observations into exact numbers. Digital processing enables to exactly measure the penetration depth of curcumin and to estimate the total amount of dermally penetrated curcumin [16].

Data obtained for the penetration depth of curcumin from the different formulations are shown in Figure 3. In comparison to the nanosuspension without additives, the penetration depth of curcumin was significantly increased when glycerol, urea or propylene glycol were added as excipients. Olive oil had no influence on the penetration depth of curcumin and ethanol decreased the penetration depth. The results are in line with the results obtained from the visual inspection and can be explained by the well-described penetration enhancing properties of glycerol, urea and propylene glycol [27,28,29,30,31,32,33,34]. Also, olive oil—due to its occlusive properties—could have been expected to yield an improved penetration depth compared to the original nanosuspension. However, the low amount that was added to the formulations might not be sufficient to yield an occlusive effect.

The decreased penetration efficacy of ethanol was not expected, because also ethanol is known to possess penetration enhancing properties [30,35,36,37]. However, it is also known to possess dehydrating properties [38]. The nanosuspension contained water and surfactant. Both compounds can increase the penetration on their own [39,40]. Therefore, it can be assumed that the skin-hydrating properties, and with this the penetration enhancing properties of water and surfactant in the nanosuspension without additives, were stronger than the penetration enhancing effect of ethanol. The addition of ethanol to this formulation might have caused a “dry-out” effect, which then reduced the swelling of the SC and with this the passive diffusion of the curcumin. The reduced thickness of the SC due to the addition of ethanol was already observed from the visual inspection of the microscopic images and could be further proven be measuring the SC thickness with ImageJ software (Figure 4).

Measuring the thickness of the SC of the untreated skin prior to the application and after 6 h of incubation revealed a decrease in the stratum corneum thickness by about 10%. This is considered to be due to a drying-out effect of the skin during the incubation time. The treatment with the nanosuspension without additives increased the stratum corneum thickness and prevented the drying of the skin during the incubation. A similar strong hydrating effect was found for the formulations containing 2% glycerol, urea or olive oil. Data confirm the observations from the microscopic images and confirm that these excipients do not alter the skin hydration when compared to the nanosuspension without excipients. In contrast, ethanol was proven to dehydrate the skin, whereas glycerol 5% and propylene glycol could further increase the skin hydration.

The determination of the penetration depth and the SC thickness gave a clear image of the influence of the different excipients on the penetration depth of curcumin and their skin hydrating properties. However, to evaluate the penetration efficacy holistically, it is also important to assess the total amount of penetrated active. In this study the amount of penetrated curcumin was assessed by analyzing the fluorescence intensity of each image after the autofluorescence of the skin was fully extracted from the original image by means of an automated threshold algorithm. Hence, the detected fluorescence intensity corresponded to the amount of penetrated curcumin [16]. Results obtained from this analysis (Figure 5) substantiate the observations from the visual inspection of the microscopic images and from the analysis of the penetration depth (Figure 2 and Figure 3). However, they are not fully in line with the results obtained from the analysis of the penetration depth, because the addition of urea was now found to be less effective than the addition of glycerol.

The differences between glycerol and urea are plausible. Water is essential to hydrate the SC and only a hydrated SC will allow for a good dermal penetration of actives. The pure application of water would shortly moisture the skin and thus allow for good penetration. Over time, the water evaporates, and the hydration of the SC will decrease. With this, the penetration efficacy decreases. Glycerol acts as humectant and prevents water loss of the stratum corneum [41,42,43]. The penetration is therefore increased compared to the nanosuspension without additives. The penetration enhancement is concentration dependent, because the increase of the concentration of glycerol, i.e., from 2% to 5%, increased the skin moisturizing properties [44].

In contrast to glycerol, which will mainly localize in the intercellular space of the SC, urea is considered to penetrate into the corneocytes which causes the corneocytes to swell [45,46,47]. Consequently, with increasing concentrations of urea, the corneocytes increase in size leading to a decrease of the intercellular space. With this, the penetration efficacy of the lipophilic curcumin, which is considered to penetrate the skin via the extracellular pathway, is decreased (Figure 6).

Data show that excipients, even though if they are used in very low concentrations, can distinctly influence the passive, dermal penetration of curcumin from nanocrystals. As a major penetration enhancing effect, a long-time hydration effect of the stratum corneum was identified, which could be achieved by adding hygroscopic excipients to the formulations. Therefore, additives that dehydrate the stratum corneum should be avoided if effective passive, dermal penetration is desired.

### 3.3. Influence on Biophysical Skin Parameters

In the next part of the study, the influence of the different excipients on the major biophysical skin parameters (TEWL, skin hydration and skin friction) was determined.

#### 3.3.1. TEWL

The TEWL is a measure for the skin barrier function. High values indicate a disrupted barrier and thus an impaired barrier function. The TEWL of the non-treated skin increased during 6 h of incubation time, indicating a slight breakdown of the SC barrier during this time. The application of the nanosuspensions decreased the TEWL, which can be explained by the formation of a film on top of the skin, which reduces water evaporation from the skin (Figure 7). The reduction in TEWL seemed to be more pronounced for the formulations that contained glycerol, urea in a concentration of 5% or 2% ethanol. This can be explained by the hygroscopic properties of glycerol and urea, which further reduce the water evaporation from the skin and from the formulation on top of the skin. The decreased TEWL of the skin treated with ethanol might be explained by the dehydrating properties of ethanol. This means, it can be assumed that ethanol and water will evaporate quickly after topical application. Thus, leading to a dry skin and to the formation of a dry formulation on top of the skin. The addition of olive oil did not alter the TEWL. However, data in this part were not significant and thus further experiments are necessary to solidify these observations.

A slight but nonsignificant increase in TEWL was seen for the skin treated with the propylene glycol containing nanosuspension. This indicates an SC impairment, which is—even though not significant in this study—well-described in the literature [27] and thus further supported by this data. The TEWL of the skin treated with the nanosuspension that contained 10% urea was increased by about 35%. The effect was significant. Hence, the barrier function-improving properties of urea which are well described in the literature [48] could not be observed in this study. A possible reason is that the barrier-protecting properties of urea are caused by the urea-induced upregulation of involucrin, filaggrin and transglutaminase-1 (TG-1) expression [48], which of course could not take place in the short-term ex-vivo model used in this study. Therefore, the barrier disrupting effect observed in this study might be explained by the keratinolytic properties of urea.

Data obtained show that the curcumin nanosuspension without excipients possesses no barrier-disruptive properties. Also, the addition of 2% or 5% glycerol, 2% olive oil, 2% ethanol or low amounts of urea (5%) did not alter the skin barrier function. Some impairment was found when 5% propylene glycol or 10% urea were added.

#### 3.3.2. Skin Hydration

Skin hydration was measured with a corneometer which determines the capacitance of a surface. As water has a high capacitance, high corneometer values correspond to a high skin hydration. After 6 h of incubation no significant changes in skin hydration were observed for the untreated control skin area (blank) and the areas treated with formulations that contained 2% glycerol, 5% urea or propylene glycol (Figure 8). All other changes were significant. The nanosuspension without additives and the formulations that contained 10% urea, ethanol, or olive oil were found to decrease the corneometer values by about 40%.

The observed decrease in capacitance is not corresponding to a real decrease in skin hydration but indicates a film formation of the formulations on top of the skin. As the formulations dry out, a lower capacitance is measured after 6 h of incubation, because the formed film is an insulator that decreases the capacitance reading. The effect has been previously observed and thus data of this study are in agreement with previous findings [49]. In contrast, the addition of 5% glycerol increased the measured skin hydration significantly. The skin-hydrating effect can be explained by the hygroscopic properties of glycerol which prevents the drying of the formulations and thus results in a higher water content in the formulation film that was formed on top of the skin. A similar but not significant trend was seen for the formulations containing 2% glycerol or 5% urea. This is reasonable because lower concentrations will possess less hygroscopic properties.

#### 3.3.3. Skin Friction

Treating the skin for 6 h with the different formulations did not significantly alter the skin friction (resistance against a rotating probe) for the nontreated control, the nanosuspension without excipients and the nanosuspensions with 10% urea or 2% ethanol. The other formulations tremendously increased the skin friction (Figure 9). Data also show that the addition of excipients led to a higher skin friction than the use of the nanosuspension without excipients. The differences are related to the different properties and interactions between formulation, skin and the rotating disc of the frictiometer. The most pronounced increase in skin friction was found after adding 2% or 5% glycerol, 5% urea or olive oil to the nanosuspension. The increased skin friction is plausible because only glycerol and 5% urea were shown to increase the skin hydration. Thus, the increased skin hydration and the swollen SC can be considered to make the skin “stickier”. Thus, leading to the higher skin friction. Likewise, propylene glycol possesses hygroscopic properties [50] and was shown to increase the thickness of the stratum corneum (c.f. Figure 2 and Figure 4). Hence, it can be considered, that propylene glycol increases the skin friction by a similar mechanism like glycerol. In contrast, olive oil can be considered to form a small, oily film on the skin. The oily film—due to the hydrophobic nature of the oil—causes a sticky and slightly greasy skin feeling, which causes an increase in the measured skin friction values.

The measurement of the biophysical skin parameters is not yet a standard procedure when testing dermal formulations in-vitro or ex-vivo. Therefore, only little scientific data are available in this regard. On the contrary, many in-vivo studies use the analysis of the biophysical skin properties as a basic tool to investigate the influence of dermal formulations on the skin conditions for different health conditions. Overcoming the missing link between in-vitro, ex-vivo and in-vivo data is thus needed to improve the power for the development and formulation of novel and efficient dermal drug products. Hence, the long-term aim of such testing is to establish a valid in-vitro / in-vivo correlation, which would then allow to forecast the in-vivo performance of topical formulations in regard to their skin care properties. If the strategy is successful, the use of skin probes in in-vitro and ex-vivo experiments can help to reduce the number of in-vivo experiments, which would not only help to reduce costs, but would also allow to reduce the number of animal experiments. With this, the proposed strategy is a straightforward approach in regard to the 3R strategy [51] and is also helpful for the development of science-based (European) cosmetic products, where animal testing is prohibited [52].

This strategy was followed in this study and included the ex-vivo testing of the three major biophysical skin properties (TEWL, skin hydration and skin friction) prior to and after application of the different formulations. Results show relatively large standard variations between the different skin areas tested. Hence, future tests should increase the number of test areas and further studies should be carried out to optimize and standardize the test procedure on the ex-vivo pig ear model, which in turn should improve the statistical strengths of such data.

The data obtained in this study already provide evidence that the determination of the biophysical skin parameters, along with the assessment of the penetration efficacy, is helpful for the selection of suitable excipients. For example, the TEWL data enabled to pinpoint formulations that tend to impair the stratum corneum, whereas corneometer values were not affected by these compounds but clearly differentiated between film forming formulations and formulations with hygroscopic—thus skin-hydrating—properties.

Whereas data that are obtained from dermal penetration testing can select the most suitable excipients for optimal passive dermal penetration, the additional testing of the biophysical skin properties can filter excipients that provide optimal skin care or skin-protective properties. In this study, the addition of 5% propylene glycol or the addition of 5% glycerol to a curcumin nanosuspension was found to be most suitable for improved passive diffusion of curcumin. However, assessing the biophysical properties that are caused on the skin due to the addition of the excipients to the nanosuspension, clearly showed that glycerol possesses skin-caring properties, whereas propylene glycol showed a trend towards barrier-impairing properties. Based on the data, glycerol could be selected to be the most suitable excipient to foster passive dermal penetration of curcumin from drug nanocrystals and to provide skin-caring properties at the same time.

### 3.4. Determination of Hair Follicle Targeting

The efficacy to target nanoparticles into the hair follicle is considered to be size-dependent [10]. As the different formulations were found to possess slightly different particle sizes that were caused due to partial dissolution of the curcumin nanocrystals upon the addition of the excipients (c.f. 3.1.), it was important to estimate the impact of these differences on the penetration efficacy of the nanocrystals into the hair follicles. This was done by determining the penetration depth of the curcumin nanocrystals without excipients directly after production (NS) and after 14 days of storage at room temperature (NS d14). During this time, the particle size increased from about 700 nm to >1 µm (LD data, d(v) 0.5). The DLS data (c.f. Table 2) indicated the existence of an additional small sized particle population that possessed a size of about 250 nm in case of the freshly prepared nanosuspension and a size of about 500 nm after 14 days storage. With this, from all formulations produced in this study, the fresh nanosuspension possessed the smallest sizes, the NS d14 the largest particle sizes and all other formulations particle sizes within this range (c.f. Table 2). Therefore, it was assumed that differences in the penetration efficacy due to differences in sizes would be most pronounced between these two formulations without excipients and less pronounced for all other formulations that contained excipients.

The mean penetration depth of the freshly prepared nanocrystals (NS) was 293 ± 103 µm and decreased by about 20% to 233 ± 63 µm for the aged nanocrystals (NS d14) with larger particle size. The difference became significant by direct comparison of the two means with the student‘s *t*-test (*p* = 0.02) but was not significant with the post-hoc tests used after one-way ANOVA. Based on the data, it was concluded that the differences in size might contribute to the penetration depth of the nanocrystals to a minor extend. Further, it could be expected that the larger sized formulations would lead to a slightly decreased penetration depth. Hence, if the size would be the most influencing parameter for the penetration of nanocrystals into the hair follicles, there should be a trend showing that the formulations possessing particle sizes close to the aged nanosuspension, i.e., the formulations that contained propylene glycol, ethanol or olive oil (c.f. Table 2), cause a reduced penetration depth in comparison to the freshly prepared nanosuspension with smaller particle size. Accordingly, the formulations that possess similarly small particle sizes than the freshly prepared nanosuspension, i.e., the formulations containing glycerol or urea (c.f. Table 2), should lead to penetration profiles being similar or close to the penetration profile of the freshly prepared nanosuspension without excipients. The results obtained disproved the expected decrease in penetration depth for the larger sized formulations and the expected similar penetration profiles for the formulations with similar particle sizes (Figure 10 and Figure 11).

Instead, a significant increase in the penetration depth was found for the formulations that contained propylene glycol and ethanol. The increase in penetration depth was about 30–35% compared to the freshly prepared nanosuspension and about 60–70% compared to the aged nanosuspension without excipients. Likewise, the addition of olive oil led to a slight increase in the penetration depth of the nanocrystals. The difference was not significant when compared to the freshly prepared nanosuspension and became significant by a direct comparison with the penetration depth of the aged nanosuspension (Mann–Whitney-test, *p* < 0.05). The addition of 5% glycerol or 10% urea decreased the penetration depth of the nanocrystals by about 15–20%. The effects were concentration dependent and became significant by a direct comparison with the freshly prepared nanosuspension (unpaired *t*-test (one-tailed), *p* < 0.05).

One-tailed tests compare mean values of two independent groups in only one direction and thus are considered to provide more power to detect whether a mean of a population is larger or smaller than the mean of another population [53]. Based on the observations in our previous study, where the addition of 5% glycerol was found to decrease the penetration depth of the curcumin nanocrystals [12], we hypothesized that also in this study the addition of glycerol would lead to a decrease in penetration depth. As we assumed that the effect occurs due to a swelling of the hair shaft and/or changes in the structure of the hair cuticula and is caused by the hygroscopic properties of the glycerol, we also hypothesized that urea and propylene glycol—which are also known to possess hygroscopic properties—will cause similar effects. Based on the significant results from the *t*-tests, the hypothesis was confirmed for glycerol and urea but was disproved for propylene glycol. The results therefore indicated that the size of the nanoparticles and the hygroscopic properties of the excipients might not be the sole parameters that affect and modulate the penetration efficacy of nanocrystals into hair follicles.

The main driving parameter for an efficient uptake of nanoparticles into the hair follicle is considered to be the ratchet effect [10,11,54]. Therefore, it was assumed that the different excipients might differently influence the hair and/or hair follicle structure which could then explain the different effects observed. In order to prove this assumption, hairs that were incubated for 6 h with the different formulations were collected and subjected to microscopic analysis (Figure 12). 

All hairs had adsorbed nanocrystals on their surface that were dried during the 6 h of incubation. The hair cuticula was clearly visible for the hairs treated with the nanocrystals without any additional excipients. However, nanocrystals were not detected within the overlapping cells of the hair cuticula when the hairs were treated with the pure nanosuspension and with the nanosuspensions that contained either glycerol or urea (Figure 12, upper row). In contrast, nanocrystals seemed to accumulate preferentially in between the cuticular structures, when the hairs were treated with propylene glycol or olive oil (Figure 12, lower row). The treatment of hairs with the nanosuspension that contained ethanol caused the formation of a crust. Due to this it was not possible to determine exactly if nanocrystals were also located in between the cuticular structures. However, the small sections that were not covered with the crust suggested this (Figure 12, lower row—middle). The observations were in line with the results obtained from the hair follicle penetration experiments and could link increased hair follicle penetration to an increased localization of the nanocrystals in between the overlapping cells of the hair cuticula, which might have caused an improved ratchet-like transport of the particles into the hair follicles.

In the next step, the hairs were rehydrated with their original dispersion medium and microscopic images were taken (Figure 13). The results showed not only differences in the adhesiveness of the nanocrystals but also differences in the structure of the hair cuticula. Almost no changes in the adherence of the nanocrystals were found for the hairs treated with the nanosuspension without excipients. The hairs treated with glycerol and urea had a smooth surface and only a few nanocrystals were found to be attached to the surface of the hair. In contrast, the re-hydrated hairs treated with propylene glycol, ethanol and olive oil showed a rougher surface of the hair cuticula and more nanocrystals were adhered to the surface of the hairs. The localization of the particles in between the cuticular structures was clearly visible and was most pronounced for the hairs treated with propylene glycol and olive oil. The results therefore further substantiate the theory that excipients can influence the penetration of nanocrystals into hair follicles due to a modification of the adhesiveness of the particles and/or due to changes in the hair cuticula structure.

The modification of the adhesiveness of the particles and/or changes in the hair cuticula structure due to the addition of excipients were further tested by simulating the application of the nanocrystals to hairs that were freshly obtained from the ventral side of different pig ears. The dry hairs were placed on a microscopic slide and 20 µL of nanosuspension with or without excipient were added. After about 5 min of incubation the hair was transferred to a clean microscopic slide and original dispersion medium was added to the hair (Figure 14 and Figure 15). The results demonstrated again clear differences in the adhesiveness of the nanocrystals between the differently treated hairs. However, the previously observed differences in the hair cuticula structure were not observed and also the adhesiveness was slightly different compared to the results obtained after 6 h of incubation. The least adhesiveness was found again for the hairs treated with glycerol. Urea led to a strong adherence of the nanocrystals, whereas ethanol and olive oil led to a formation of a crust. The crust was washed off upon redispersion of the hair in case of ethanol (Figure 14, black arrow) but remained on the hair surface in case of the hair treated with olive oil. Digital zooming of the images taken indicates that the nanocrystals are homogeneously distributed around the hair (Figure 15). Nonetheless, a detailed inspection shows that the particles start to localize in between the overlapping structures of the hair cuticula (Figure 15, areas marked with arrows). The results therefore suggest that changes in the hair cuticula structure occur not instantly and that the adhesiveness of the particles might also change over time. However, based on the assumption from our last study where we concluded that most of the nanocrystals are transported into the hair follicles within the first minutes after the application [12], the effects that were observed from the last experiment should be considered to be the main effects that cause the differences in the penetration depth of the nanocrystals.

All the results obtained in the last part of this study lead to a conclusion that the addition of excipients can alter distinctly the penetration efficacy of nanocrystals into hair follicles. The concrete mechanisms that caused the increase or decrease in penetration depth when compared to the nanocrystals without excipients could not be finally ascertained from the data obtained in this study. Nonetheless, the data suggest that the mechanism by which particles are transported into hair follicle is not only influenced by the particle size but also by some additional parameters.

One parameter that was identified in this study is the adhesiveness of the nanocrystals to the hairs. Particles that stick too tightly to the hair cannot move and—consequently—will remain on top of the surface of the hair and cannot penetrate into the hair follicle. Particles that possess extremely poor adhesiveness to the hair or are even repelled from the hairs, cannot enter the ratchet-mechanism, and thus will also not penetrate into the hair follicle. This means particles that penetrate well into the hair follicle need to adhere to the hair loosely and should preferentially locate in between the gaps of the overlapping cells of the hair cuticula to enter the ratchet mechanism. It can be speculated that excipients that modify the structure of the hair cuticula might increase the ratchet effect if they increase the roughness of the hair cuticula, whereas excipients that reduce the roughness of the hair surface will decrease the ratchet effect. Other mechanisms that promote or impede the adhesiveness of particles to hairs might be related to the charge of the hair and/or the particles and to other attractive or repulsive forces that need to be elucidated in future studies.

A second parameter that might affect the penetration efficacy of nanocrystals into the hair follicles is the condition of the hair follicle, which acts as the “pawl” and is thus indispensable as “countermovement” for a functioning ratchet mechanism and an efficient transport of the particles into the hair follicles. Consequently, it can be hypothesized that excipients that modify the conditions of the hair follicle, might also modify the “pawl” and with this the penetration efficacy of particles into the hair follicles. Possible changes that can occur involve changes in the structure of the hair follicle stratum corneum, which could become softer or more rigid and could swell or shrink. This could further modify the grip and/or the distance between the ratchet and the pawl leading to changes in the penetration efficacy.

A third condition to be considered to change upon the addition of excipients is the fluidity of the sebum. The upper part of the hair follicle, i.e., the infundibulum, is located between the duct of the sebaceous gland and the stratum corneum surface [53,54,55] and is filled with lipophilic viscous sebum that consists of neutral and nonpolar lipids [55,56,57]. A high viscosity of the sebum can be considered to counteract the movement of the ratchet-pawl mechanism. Hence, it can be assumed that a decreased sebum viscosity can accelerate the ratchet-pawl mechanism and with this, the penetration efficacy of particles into the hair follicles. On the contrary, compounds or conditions that increase the viscosity should then decrease the penetration efficacy.

Based on these considerations, the penetration of particles into hair follicles can be assumed to be a three-step process. In the first step, the particles need to adhere loosely to the surface of the hair. From there—in the second step—they need to locate in between the overlapping cells of the hair cuticula to enter the ratchet-mechanism. Finally, in the third step, the ratchet together with the pawl will then transport the particles into the hair follicle. The particle size can be assumed to mainly influence the efficacy of the particles to locate in between the overlapping cells of the hair cuticula [10]. Based on the findings of this study, it was shown that excipients can “overwrite” the size effect of the particles. In addition, they can be considered to enable a modification of all three steps of the hair follicle penetration process. This means that not only the production of tailor-made particles but mainly a purposeful selection of the excipients can help to improve or to impede the penetration of particles into hair follicles. Both aspects might be interesting in formulation development of topical products. Impeded penetration of particles might be interesting to avoid excessive uptake of unwanted particles, for example sun blockers, e.g., titanium dioxide particles [58]. Improved uptake is highly interesting for drug delivery.

The excipients used in this study were found to lead to different effects and can be considered to influence the penetration process of the particles at different stages. Glycerol reduced the adhesiveness of the particles and thus was found to reduce the penetration efficacy of the nanocrystals. If the effect was due to a decrease in the roughness of the hair cuticula and if the effect was further boosted by a change in the hair follicle surface could not be demonstrated in this study. Urea was also found to decrease the penetration depth of the nanocrystals. However, the mechanism that hampered the penetration is probably not like that of glycerol. Urea has keratinolytic properties [59,60,61,62]. After a short incubation time this caused a strong adhesion of the particles to the surface of the hairs. Over time, keratolysis resulted in a smoother hair structure (c.f. Figure 13) and low adhesiveness of particles to the hair surface. Both effects can be regarded to contribute to the observed decrease in the hair follicle penetration efficacy. Propylene glycol, ethanol and olive oil increased the penetration efficacy of the nanocrystals. As all three excipients can act as solvents for polar, lipophilic compounds [63,64,65] these excipients can also be considered to interact with the sebum and to decrease its viscosity [66,67], which then leads to an improved ratchet-effect and a resulting increase in the penetration efficacy. The more pronounced increase in penetration depth for ethanol and propylene glycol might be explained by a slight increase in hair roughness (c.f. Figure 13), which increased the space between the overlapping cells of the hair cuticula and thus allowed more particles to locate in between these cells which than allowed for an increased penetration efficacy of the particles into the hair follicles.

The data obtained for the interpretation of the different penetration mechanisms are yet too preliminary to draw a final conclusion and more research is needed in this regard. However, findings so far suggest that the addition of 2% ethanol or 5% propylene glycol to a nanosuspension are useful to enhance the uptake of nanocrystals into hair follicles. Depending on the desired route of administration, ethanol should be selected for effective hair follicle targeting without intense passive dermal penetration and propylene glycol should be used for formulations that should allow for hair follicle targeting with intense passive dermal penetration at the same time.

## 4. Conclusions

This study investigated the influence of different excipients on the penetration efficacy of curcumin nanocrystals into the hair follicles and on the passive, dermal penetration of dissolved curcumin from nanocrystals. The results revealed that the excipients influenced the penetration pathway tremendously. The addition of glycerol to nanocrystals impaired the hair follicle penetration and improved the passive, dermal penetration. Ethanol significantly enhanced the follicular penetration efficacy and reduced the passive penetration of the active at the same time. Propylene glycol improved both penetration routes. Thus, the addition of different components to the nanosuspension could define the penetration pathway of nanocrystals. The different effects of the various excipients on the passive, dermal penetration of curcumin are related to their different mechanisms of actions and are in line with the current scientific opinion about the different penetration enhancing properties of the different excipients, whereas the effects of excipients on the efficacy of the hair follicle targeting seemed to overwrite the previously published size effect of particles.

The effects observed can be explained by a three-step mechanism that transports nanocrystals into the hair follicle. Each step of this mechanism can be altered or modified upon the addition of excipients, thus leading to increased or decreased penetration of particles into the hair follicles (Table 3).

The findings of the present study suggest that fixed combinations of nanocrystals and selected excipients can be used for targeted dermal drug delivery. This means one can choose if the active compound should be delivered via passive diffusion and/or as drug reservoir by depositing active compounds such as drug nanocrystals into the hair follicles. With this, the findings provide a new perspective for the formulation of highly effective topical products.

## Figures and Tables

**Figure 1 nanomaterials-10-02323-f001:**
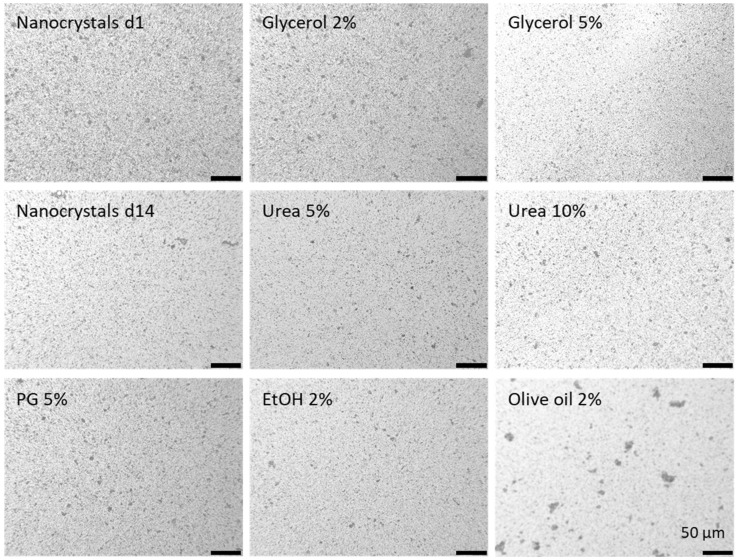
Microscopic images of the curcumin nanosuspension at the day of production, after 14 days of storage and after the addition of different excipients. Magnification: 400-fold.

**Figure 2 nanomaterials-10-02323-f002:**
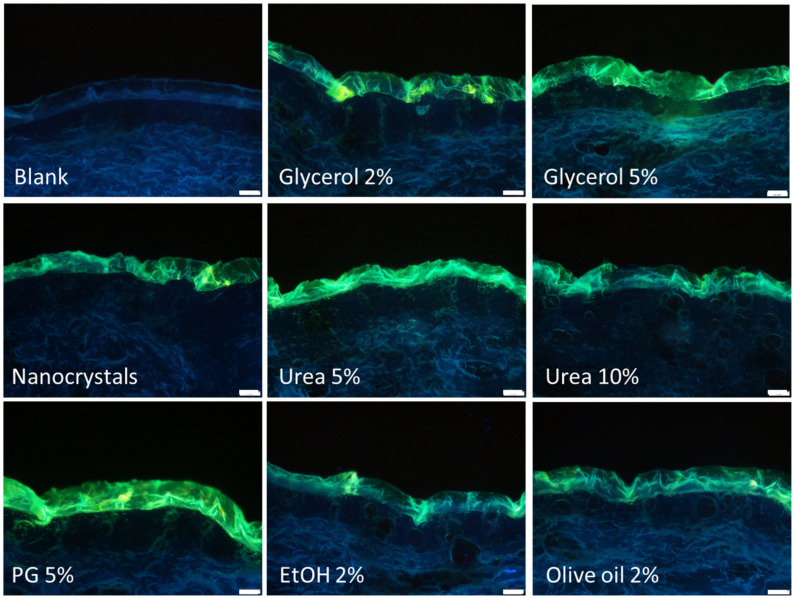
Influence of excipients on passive, dermal penetration of curcumin from drug nanocrystals. Representative images from vertical skin sections. Non-treated (Blank), treated with aqueous nanosuspension without further excipients and treated with curcumin nanosuspensions that contained different excipients, i.e., glycerol 2% and 5%, urea 5% and 10%, propylene glycol 5%, ethanol 2% or olive oil 2%. For more detailed inspection of the data, please refer to the Appendix A.

**Figure 3 nanomaterials-10-02323-f003:**
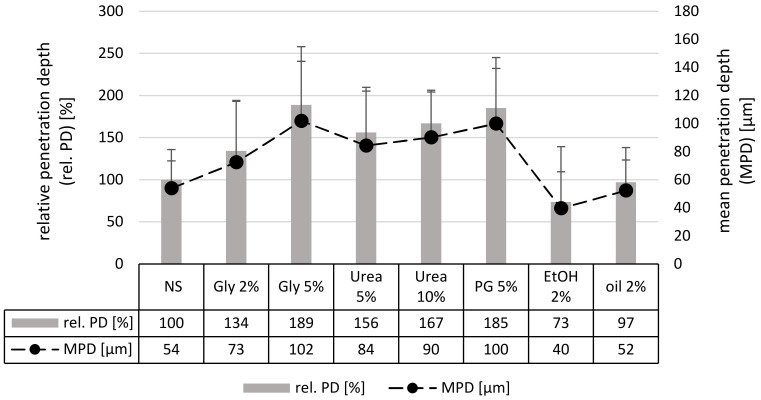
Influence of excipients on the penetration depth of curcumin determined by digital image analysis. MPD = mean penetration depth [µm], rel. PD = relative penetration depth [%] in comparison to the penetration depth of curcumin from the nanosuspension without additives.

**Figure 4 nanomaterials-10-02323-f004:**
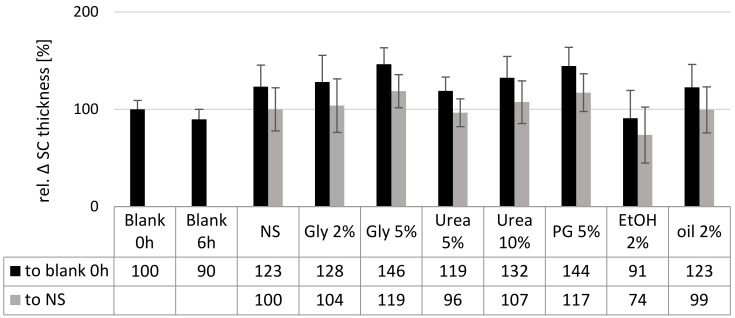
Influence of excipients on the stratum corneum thickness determined by digital image analysis. Black columns: relative thickness of SC when compared to non-treated, fresh skin. Grey columns: relative thickness of SC when compared to the SC of the skin treated with aqueous curcumin nanosuspension.

**Figure 5 nanomaterials-10-02323-f005:**
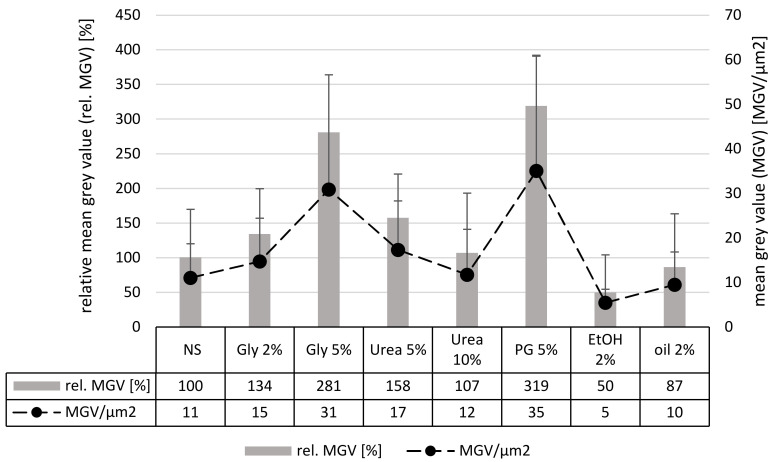
Influence of excipients on the penetration efficacy of curcumin determined by digital image analysis. MGV/µm^2^ = mean grey value per µm^2^ skin—representing the total amount of penetrated curcumin; rel. MGV [%] = relative amount of penetrated curcumin in comparison to the MGV from the nanosuspension without additives.

**Figure 6 nanomaterials-10-02323-f006:**
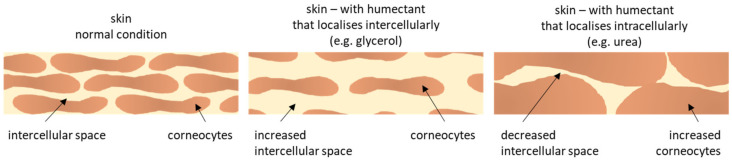
Principle of penetration enhancing properties of different humectants.

**Figure 7 nanomaterials-10-02323-f007:**
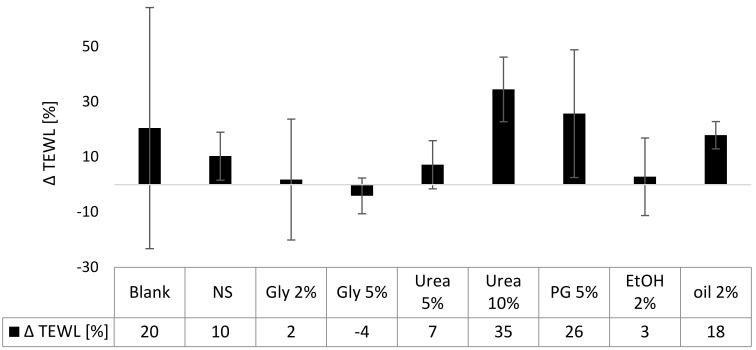
Influence of curcumin nanosuspension without (NS) and with different excipients on the skin barrier function (TEWL measurements are expressed as Δ TEWL [%], representing the relative changes in TEWL 6 h after the treatment, compared to the TEWL values prior to the skin treatments).

**Figure 8 nanomaterials-10-02323-f008:**
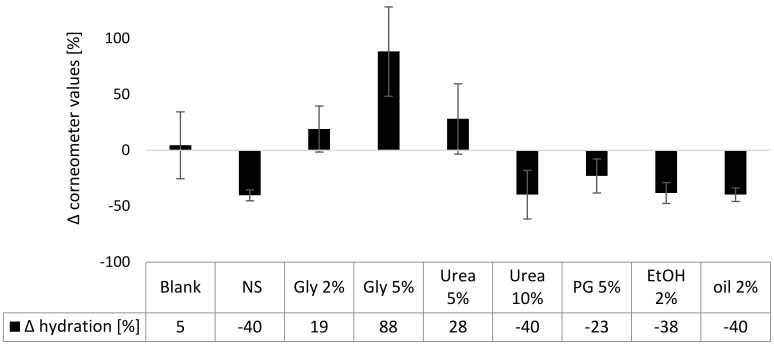
Influence of curcumin nanosuspension without (NS) and with different excipients on the skin hydration (Corneometer measurements. The values obtained after 6 h of incubation are expressed as relative changes compared to the untreated skin prior to the application).

**Figure 9 nanomaterials-10-02323-f009:**
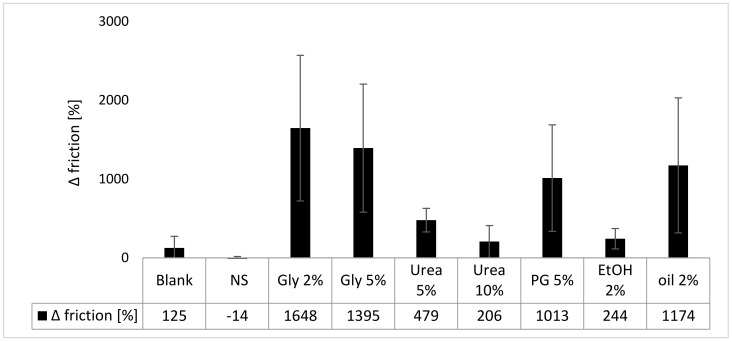
Influence of curcumin nanosuspension without (NS) and with different excipients on the skin friction (The values obtained after 6 h of incubation are expressed as relative changes compared to the untreated skin prior to the application).

**Figure 10 nanomaterials-10-02323-f010:**
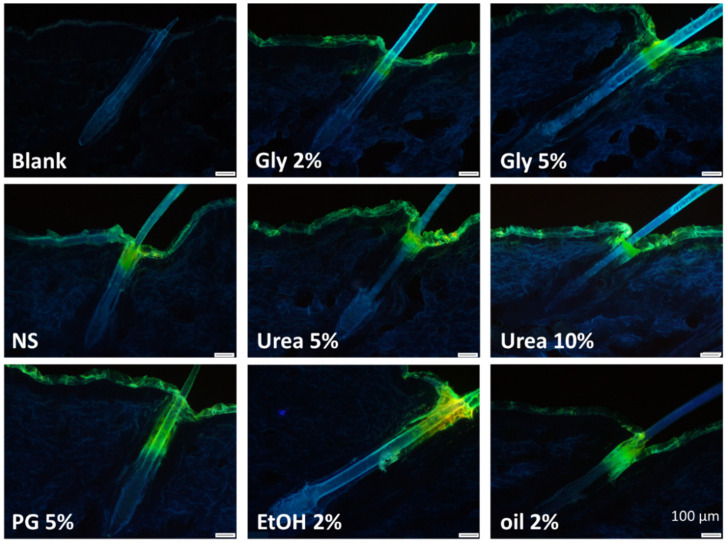
Images obtained from the skin sections (magnification 100-fold) to visualize the influence of excipients on the penetration depth of curcumin nanocrystals into hair follicles. Upper: original images. Lower: images after digital processing (transfer into black-white images and contrast enhancement) to improve the visibility of the penetrated curcumin nanocrystals. Penetrated curcumin nanocrystals appear black (indicated with black arrows). For more detailed inspection of the data, please refer to the Appendix A.

**Figure 11 nanomaterials-10-02323-f011:**
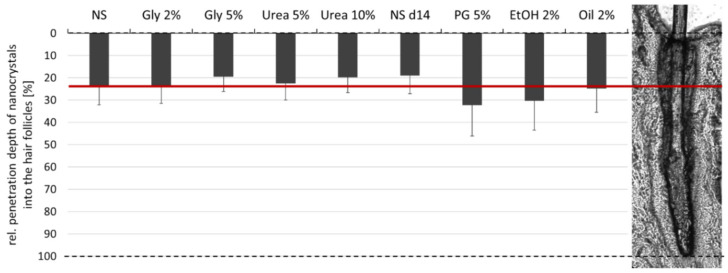
Relative penetration depth [%] of curcumin nanocrystals into the hair follicles from formulations containing different types and amounts of excipients. The relative penetration depth was calculated from the determined mean penetration depth obtained from each formulation and was related to the mean lengths of the hair follicles (porcine ear—ventral side) that was determined to be 1222 µm.

**Figure 12 nanomaterials-10-02323-f012:**
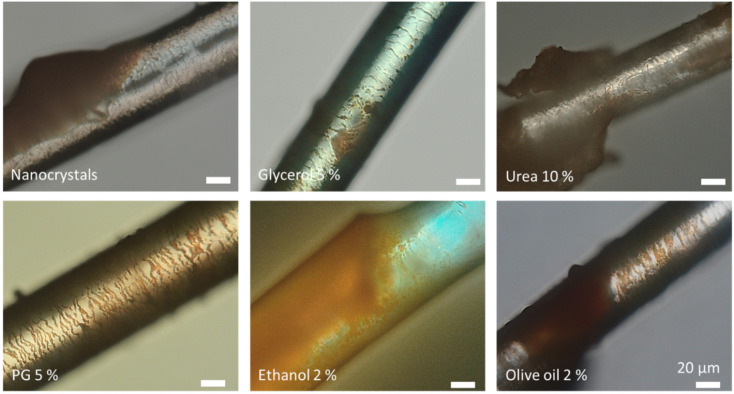
Microscopic images of hairs removed from porcine ears treated with nanosuspensions without and with excipients after 6 h of incubation (digital zoom of images taken at 400-fold magnification).

**Figure 13 nanomaterials-10-02323-f013:**
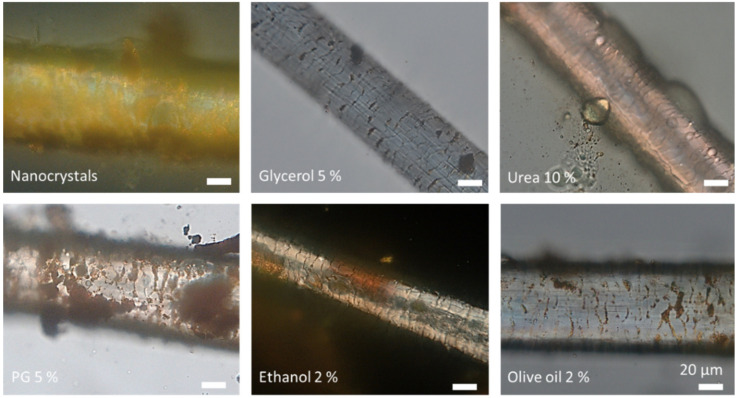
Microscopic images of hairs removed from porcine ears treated with nanosuspensions without and with excipients after addition of dispersion medium (digital zoom of images taken at 400-fold magnification).

**Figure 14 nanomaterials-10-02323-f014:**
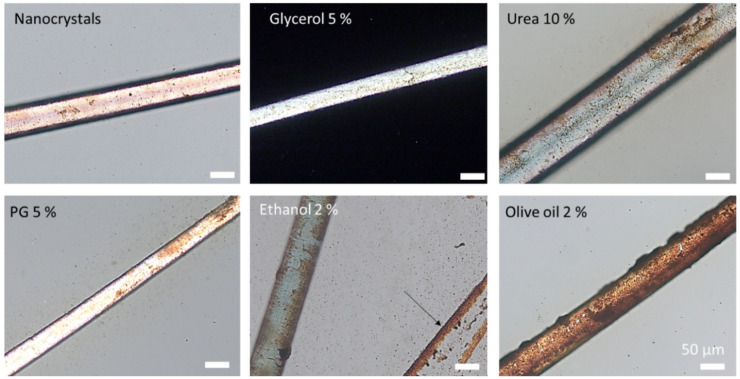
Microscopic images of porcine hairs incubated with nanosuspensions without and with excipients for about 5 min and after subsequent redispersion in the respective dispersion medium (digital zoom of images taken at 200-fold magnification).

**Figure 15 nanomaterials-10-02323-f015:**
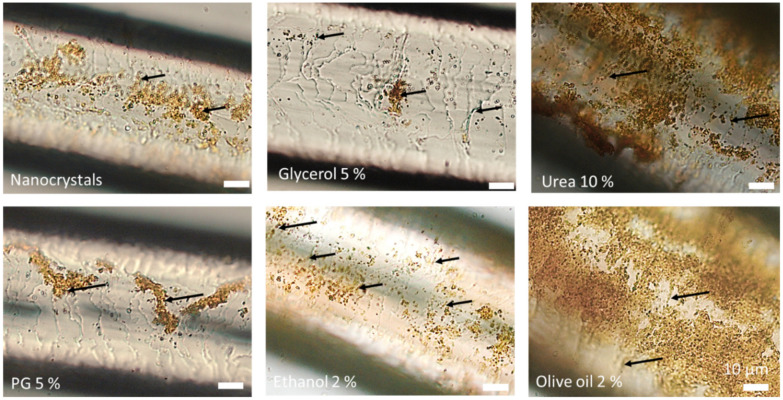
Microscopic images of porcine hairs incubated with nanosuspensions without and with excipients for about 5 min and subsequent re-dispersion in the respective dispersion medium (digital zoom of images taken at 1000-fold magnification).

**Table 1 nanomaterials-10-02323-t001:** Overview of formulations produced in this study.

Formulation Code	Type of Excipient	Concentration of Excipient (w/w)
NS	-	-
Gly 2%	glycerol	2%
Gly 5%	glycerol	5%
Urea 5%	urea	5%
Urea 10%	urea	10%
PG 5%	propylene glycol	5%
EtOH 2%	ethanol	2%
oil 2%	olive oil	2%

**Table 2 nanomaterials-10-02323-t002:** Particle sizes and size distributions (DLS data and LD data) for the curcumin bulk material, the nanosuspension without excipients and for the nanosuspensions after the addition of different excipients.

Formulation	DLS Data						LD Data											
	z-average [nm]			PDI			d(v) 0.5 [µm]			d(v) 0.9 [µm]			d(v) 0.95 [µm]			d(v) 0.99 [µm]		
bulk material	n.a.						22.6	±	0.2	50.3	±	0.5	60.1	±	0.9	77.3	±	1.7
NS d1	253	±	6	0.25	±	0.05	0.7	±	0.0	4.2	±	0.1	5.5	±	0.1	8.2	±	0.2
NS d14	483	±	75	0.48	±	0.05	1.1	±	0.0	4.8	±	0.1	6.5	±	0.1	10.6	±	0.2
Gly 2%	325	±	10	0.25	±	0.03	0.7	±	0.0	4.3	±	0.1	5.6	±	0.1	8.4	±	0.1
Gly 5%	278	±	7	0.30	±	0.03	0.8	±	0.0	4.4	±	0.1	5.7	±	0.1	8.5	±	0.1
Urea 5%	270	±	7	0.24	±	0.05	0.5	±	0.0	4.0	±	0.0	5.2	±	0.0	7.7	±	0.1
Urea 10%	277	±	7	0.29	±	0.02	0.4	±	0.0	3.8	±	0.0	5.0	±	0.0	7.5	±	0.1
PG 5%	389	±	47	0.39	±	0.07	1.1	±	0.0	4.6	±	0.1	6.0	±	0.2	8.9	±	0.3
EtOH 2%	389	±	37	0.45	±	0.09	1.1	±	0.0	4.7	±	0.1	6.1	±	0.1	9.4	±	0.2
oil 2%	459	±	169	0.54	±	0.25	1.4	±	0.1	8.5	±	0.9	13.0	±	1.6	27.0	±	3.6

**Table 3 nanomaterials-10-02323-t003:** Overview of influence of excipients on passive dermal penetration and hair follicle targeting.

Type and Concentration of Excipient	Effect on Passive Dermal Diffusion	Effect on Hair Follicle Targeting
Glycerol 2%	↑	↔
Glycerol 5% *	↑↑↑↑	↓
Urea 5%	↑↑	↔
Urea 10%	↑	↓
Propylene glycol 5% **	↑↑↑↑	↑↑↑
EtOH 2% ***	↓	↑↑↑
Olive oil 2%	↔	↑

* Suggested for drug delivery via passive dermal penetration without depot effect. ** Suggested drug delivery via passive dermal penetration with simultaneous depot effect. *** Suggested drug delivery via hair follicles without passive dermal penetration.

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
