# Peer review of "Hair Follicle Targeting and Dermal Drug Delivery with Curcumin Drug Nanocrystals—Essential Influence of Excipients"

_nanomaterials, 2020, doi:10.3390/nano10112323_

Round 1
Reviewer 1 Report
Dear the authors...
The authors reported regarding “Hair follicle targting and dermal drug delivery with drug nanocrystals-essential influence of excipients”
The manuscript contained many data with the carefull discussions from the authors.
However I think this manuscript requiring carefull descriptions for the successful publication to “nanomaterials’ due to the following reasons
[Major comments]
1. In the title of the manuscript, it is described that the hair follicle is targeted using nanocrystals. However, the title does not state which drugs are used.
2. In addition, the manuscript also said that it targets the hair follicle using nanocrystal, but it is difficult to say that the drug was targeted to the hair follicle because it was moved by simple diffusion, skin hydration and so on.
3. In general, research methods, results, and conclusions should be described in abstract, but very broad content is described, and it is difficult to regard it as abstract.
4. The authors used glycerol, urea, propyleneglycol, EtOH, and olive oil for dermal delivery and hair follicle targeting of curcumin. We believe that at least a solubility test should have been performed to see the effects of these additives.
5. In Table 2, it is strange that the difference between the DLS and LD results is more than doubled. The authors explained that this phenomenon is the result of Ostwald repening, but I think this phenomenon can be said after a long period of observation.
6. In Lines 234~236, the authors described that the solubility of curcumin was very good in EtOH and propylene glycol, and that the particles were the largest in the forlumation containing EtOH and propyleneglycol. However, the higher the solubility, the smaller the particle.
7. In section 3.2.1., the authors reviewed and did not cite any references. For the discussions to be meaningful, comparison with existing literature is essential. This is not just for this part, it is something that is found throughout the discussion parts.
8. I think Figure 10 is one of the most important data in this manuscript. Contrary to what the author emphasized in the title and in the text, curcumin was neither delivered to the hair follicle nor was 'targeted'.
9. What is the'ratchet-mechanism' in line 615?
10. As mentioned above 3.3.4. Determination of hair follicle targeting was described in pp. 14-20, no single reference was cited. It is advisable to describe the discussion through comparison with existing literature.
11. "4. Conclusion" of p20 is too long. Please describe in a conclusive manner.
12. There are many grammatical errors in the manuscript. There are many things to point out about the overall content, so I didn't point out one by one. In order for the manuscript to be published in 'nanomaterials' JOURNAL, it seems necessary to have English proofreading by a native speaker.
[Minor comments]
1. Please write ‘Ⓡ’ in superscript.
2. Replace olive oil with olive oil in Table 1
3. Please unify the way the legends are expressed in Figure 4 and Figure 5. For example, unify whether to use Gly2% or 2% Gly.
4. In Figures 7 and 8, when the deviation bar is minus, it is thought that it should be drawn downward. It seems to be drawn with the Exel program. Please try using more powerful software.
Author Response
Thank you very much for your time and your comments. Please see pdf file below:

Reviewer 2 Report
This study describes the preparation of curcumin nanocrystals with various excipients and investigates the impact of these excipients on the passive dermal penetration and hair follicle targeting of curcumin. Moreover, their influence on bio-physical skin properties, including skin barrier function, skin hydration and skin friction, has been thoroughly characterized. The conclusion is well supported by a sufficient body of solid data. This manuscript requires only minor revisions, as follows:
1] Figure 3 shows the influence of excipients on penetration depth of curcumin. Although this figure shows two different data (mean penetration depth and relative penetration depth), there is only one Y-axis named “mean penetration depth” on the left side. To avoid any confusion, it’s recommended to add another Y-axis representing “relative penetration depth” to the right side.
2] Similarly, Figure 5 shows two different data (mean grey value (MGV) and relative MGV), but there is only one Y-axis named “mean grey value”. Please add another Y-axis representing “relative MGV” to the right side.
Author Response
We thank the reviewer for his comments. We added a second Y-axe to Fig. 3 and 5 respectively to avoid such confusion.
In addition, please find all our answers to the reviewers comments in the pdf file attached.

Reviewer 3 Report
The assessed paper deals with ithe issue of active pharmaceutical ingredients (API) delivery through the skin. In my opinion, the paper needs major revision to rove the relevant features of the used "nanomaterials" (declared by the authors). Below are the major concorns:
- The curcumin nanocrystals are used as model API. Table 1 shows LS results which elcidates sizes of over 300 nm. The z-average dimension is much over 100 nm (suitable in Nanomaterials journal). Is there any truly nano-fraction or just sub-micron sizes? If only sub-micron than this study is not suitable for this journal.
- Fig. 1 shows further characterisation of the API and this is light microscope (not SEM or TEM!). The Magnification is only 400fold. Therefore, the tested particles are not nano-sized.
- To detail the structure of the model API, SEM and TEM images are strongly needed. Also, XRD characterisation, FTIR and also UV-VIs to verify the absorption signals (they change upon size)
- Nevertheless, the skin-related testing is done in a very high and thorough manner with deep discussion on mechanisms of action. On the other hand, if mentioned claims and discussions are not supported with thorough material analysis, they are vague and not convincing.
Author Response
Thank you very much for your time and your comments. Please see pdf file attached.
Round 2
Reviewer 1 Report
Reviewer 1 comments:
It still containing mistakes and not fine tunned and I'd like to decide 'reject' for the publication to 'Nanomaterials' journal for the following reasons.
The main reasons for reject are...
The data presentation is still not scientific. For example
-The didn't suggest original particle size of curcumin before and after milling process.
-The instrumental analysis using SEM or TEM would be required for characterization of curcumin
nanocrystal.
- The did statistical analysis however no statistical analysis was conducted and commented elswhere.
- Authors used 'penetration' however no data suggested 'penetration effect' of nanocrystal and
excipients through hair follicles and dermis.
The abstract is not abstractive. It shoud address simply the purpose, experiemental process main
rsults and conclusion.
Purpose is too long and methods were not described and results if very broad and no conclusion(not
conclusive).
I also suggest minor comments
- Line 160, 199, 201, 204... : ® should be superscript
- Line 522 : (Fig. 10, 11) -> (Fig. 10 and 11)
- Line 529 : (Mann-Whitney-test, p = 0.01) -> (Mann-Whitney-test, /p/ < 0.01)
- Line 532 : p-values < 0.05 -> p < 0.05
- What's the meaning of '↔' in Table 3?
The authors insisted that it was found from another articles, I suggest to change 'hair follicular
targeting' as 'hair follicular delivery'.
Initially, it was intended to be corrected by a native speaker.
However, the manuscript doesn't seem to have gone through the proofreading process by native
speakers.
English expressions are required proofreading, and we recommend that authors get proofreading
from a native speaker.
Author Response
See attached document

Round 3
Reviewer 1 Report
Dear the authors..
The authors changed following the revision comments by the reviewer.
I think the changed parts looks acceptable for the publication to "nanomaterials"
Sincerely yours
Author Response
We thank the reviewer for his time and engagement to review the manuscript.